# Real-Time Person Identification in a Hospital Setting: A Systematic Review

**DOI:** 10.3390/s20143937

**Published:** 2020-07-15

**Authors:** Heleen M. Essink, Armelle Knops, Amber M.A. Liqui Lung, C. Nienke van der Meulen, Nino L. Wouters, Aart J. van der Molen, Wouter J.H. Veldkamp, M. Frank Termaat

**Affiliations:** 1Department of Clinical Technology, Faculty of Mechanical, Maritime and Materials Engineering (3Me), Delft University of Technology, Mekelweg 2, 2628 CD Delft, The Netherlands; heleenessink@gmail.com (H.M.E.); armelle.knops@gmail.com (A.K.); amberliquilung@live.nl (A.M.A.L.L.); nienkemeulen@gmail.com (C.N.v.d.M.); 2Department of Radiology, Albinusdreef 2, NL-2333 ZA Leiden, The Netherlands; A.J.van_der_Molen@lumc.nl (A.J.v.d.M.); w.j.h.veldkamp@lumc.nl (W.J.H.V.); 3Department of Surgery, Albinusdreef 2, NL-2333 ZA Leiden, The Netherlands; m.f.termaat@lumc.nl

**Keywords:** real-time person identification, RFID, biometric identification, hospital, healthcare

## Abstract

In the critical setting of a trauma team activation, team composition is crucial information that should be accessible at a glance. This calls for a technological solution, which are widely available, that allows access to the whereabouts of personnel. This diversity presents decision makers and users with many choices and considerations. The aim of this review is to give a comprehensive overview of available real-time person identification techniques and their respective characteristics. A systematic literature review was performed to create an overview of identification techniques that have been tested in medical settings or already have been implemented in clinical practice. These techniques have been investigated on a total of seven characteristics: costs, usability, accuracy, response time, hygiene, privacy, and user safety. The search was performed on 11 May 2020 in PubMed and the Web of Science Core Collection. PubMed and Web of Science yielded a total n = 265 and n = 228 records, respectively. The review process resulted in n = 23 included records. A total of seven techniques were identified: (a) active and (b) passive Radio-Frequency Identification (RFID) based systems, (c) fingerprint, (d) iris, and (e) facial identification systems and infrared (IR) (f) and ultrasound (US) (g) based systems. Active RFID was largely documented in the included literature. Only a few could be found about the passive systems. Biometric (c, d, and e) technologies were described in a variety of applications. IR and US techniques appeared to be a niche, as they were only spoken of in few (n = 3) studies.

## 1. Introduction

Acute trauma care for severely injured patients is performed by a multi-disciplinary team of in-hospital specialists. The team takes care of every major trauma patient presented to a trauma center 24/7 and is activated within minutes after announcement. A trauma team activation is a critical time-sensitive procedure where communication is vital [1,2,3]. Miscommunication is one of the big factors that can lead to an unwanted patient outcome [4,5]. Knowledge of the team composition is the basis for good communication within a team [6]. This is a challenge during acute trauma care, since the team composition differs daily and consists of a variety of disciplines [1,3]. Therefore, the identification of caregivers, to create a real-time overview of, for example, the name and function of the present caregivers at the trauma room would be useful.

Currently there are many techniques allowing for real-time person identification in healthcare [7], defined in this review as being able to identify a person at any given time. These techniques have already been implemented in different parts of the healthcare system, ranging from patient tracking [8] in hospitals to physicians’ attendance [9]. Each technique, from Near-Field Communication (NFC) devices [10] to WiFi based systems [11], has different characteristics that make it suitable or not for specific applications. A project was initiated with the aim to design a system that would allow a real-time overview of present healthcare workers and team completeness during a trauma team activation. To achieve this design, a system to identify healthcare workers had to be chosen. This systematic literature review was performed to support the choice of such a system. In this trauma setting, where everything is mission critical, costs, accuracy, and speed are essential characteristics. Furthermore, the usability of the technologies is a context-specific aspect that has to be accounted for [12]. Many of these different characteristics have been investigated for the currently existing technologies. The diversity of the available technologies together with the number of aspects that have to be accounted for calls for a comprehensive overview. The aim of this qualitative systematic review is to assess the different types of real-time person identification available in healthcare and investigate their characteristics regarding costs, usability, accuracy, response time, hygiene, privacy, and user safety.

## 2. Method

We performed two systematic electronic searches in PubMed and Web of Science according to the Preferred Reporting Items for Systematic Reviews and Meta-Analyses (PRISMA) statement [13]. The final search in these databases was performed on 11 May 2020. Removal of duplicates within the retrieved articles in the two databases was performed in EndNote X9 (Clarivate Analytics, Philadelphia, PA, USA) [14].

In line with the research question, a broad database search was conducted. The most important terms in our search string ([majr]) were “Radio Frequency Identification Device”, “Biometric Identification”, “Costs and Cost Analysis”, “Privacy”, “Safety”, “Safety Management”, “Equipment Safety”, “Hygiene”, “Infections”, “Dimensional Measurement Accuracy”, “Data Accuracy”, “Sensitivity and Specificity”, “Hospitals”, and “Health Care Category”. Search terms for identification using cards was added as text word field tags ([tw]). Search terms covering the outcomes of response time and usability were added as title field tags ([ti]) (for the full search strategy, see Appendix A). We used the setting “most recent” in PubMed. In both databases, we only searched for articles from the last ten years because we wanted to gain insight into the most recent technologies.

An article was included when it (1) described a technical solution or system for the identification of patients or healthcare personnel (2.a) that was currently being used in a medical setting (hospital, private clinic, or global health) or (2.b) for which the aim was to be used in a medical setting and (3) gave information on at least one of the outcome measures (costs, usability, accuracy, response time, hygiene, privacy, or user safety) regarding the identification component of the technique or system. Furthermore, the outcome measures were to be retrieved after (4.a) an implementation in healthcare practice or (4.b) a test of a prototype or proof of concept, in a test setting adequately simulating the medical setting and a medical procedure in which the identification method would be used. We only included articles describing the identification of living people by using non-invasive identification methods. We made the assumption that all studies describing Real-Time Location Tracking Systems (RTLS) used the identification of living persons even when this was not explicitly described, since a location could not be assigned to someone without identifying the person. We excluded the implementation of identification in out-hospital elderly care settings. Articles using identification based on DNA tests, X-ray, Computer Tomography (CT), and Magnetic Resonance Imaging (MRI) were also excluded. Articles describing surveys, regarding the overall use of Radio Frequency Identification Devices (RFID) in different hospitals, were excluded if their results could not be reduced to the individual applications of the technique. Lastly, reviews or articles that were not written in English were excluded.

Five investigators (HME, AK, AMALL, CNvdM, NLW), independently, screened the titles and abstracts of the citations for whether they met the inclusion criteria and if they were not in conflict with the exclusion criteria. Every title and abstract had at least been screened by two of the five investigators. When the investigators did not agree with each other, the article was included for full-text analysis. Full-text studies were divided over the same investigators. Every investigator received different articles than for the abstract analysis. Individual full-text inclusions and exclusions were presented to the other four investigators. Reviewers resolved discrepancies through discussion, and full-text selections were merged to one final set of included articles for this systematic review.

Assessment of quality and bias was conducted by using scales that were composed by some of the authors (AK, CNvdM, and HME). These scales are described in Appendix B. Data of the included studies were extracted on a data extraction form made by the authors including technical details of the identification technique, the identified subjects (patients and/or personnel), the (aimed) implementation setting and the goal of the article, the described outcome measures, the method on how the information on these outcome measures had been retrieved, and the results of the outcome measures.

## 3. Results

A total of n = 265 records were identified through PubMed and n = 228 records through Clarivate Web of Science v5.35 (Figure 1). After duplicate removal (n = 29), a total of 464 records were screened on title and abstract, and 379 were excluded. The remaining 85 records were scored on full text. A total of 59 records were excluded based on the following criteria: not implemented or tested in a medical setting (n = 26), forensic science (n = 1), reviews (n = 2), no outcomes related to identification (n = 25), used in elderly care or geriatrics (n = 5), or asset tracking (n = 3). The review process finally resulted in a total of n = 23 articles.

The 23 included records described a total of seven distinctive techniques (Table 1). The first and most common technique was active RFID tags (n = 13) (Table 2); RFID covers a large portion of the electromagnetic frequency spectrum (120 MHz up to 10 GHz). The second technique was a variation on the aforementioned one: passive RFID tags (n = 1) (Table 2). A few (n = 3) records did not specify which type of RFID technique was employed (Table 2). The second group was biometric identification techniques (Table 3), including fingerprint (n = 3), facial recognition (n = 1), and iris identification (n = 1). The last two techniques that were retrieved (Table 4) were ultrasound (US, n = 1) and infrared (IR, n = 2) tracking.

### 3.1. RFID

#### 3.1.1. Active RFID

Active RFID tags were the most commonly encountered group in this review (n = 13). Three different types of techniques were identified (Table 2): WiFi based (2.4 GHz and 5 GHz, n = 2), Bluetooth based (2.400 GHz and 2.483 GHz, n = 4) and proprietary bands (between the indicated 120 MHz and 10 GHz, n = 7).

Location accuracy for most systems were between 1 and 4 m [21,30]. RFID was shown to register entrance and departure times accurately in rooms [20]. This same study [20], among others [16,25,26], reported a problem with false positives. Nearby badges were registered while the person was not physically present in the room. Other factors influenced accuracy as well: one study reported a 10% rise in accuracy when using two, instead of one, antennas in the same room [25]. RFID tags not visible on patients compromised the accuracy of identification [29]. Tags in a wrist band could only be detected in a 5 cm proximity to the detector, while tags on an ID card were detected from a distance of 80 cm [32]. Furthermore, overcrowded rooms caused interference between the tags [18]. Overall, accuracy varied largely between studies due to the different scenarios of implementation, user skills, and knowledge. The lowest reported accuracy of identification was 52.4% [31]; this was reported in a clinical setting, which was significantly lower than the 85% of RFID tags being correctly identified in the test setting. Besides this value, accuracy of identification varied from 82% (ICU department) to 100% (six persons in one OR room) [17,24,25].

Implementation in both the workplace and the workflow were time-consuming processes that asked for continuous feedback from the end-user [17,30]. WiFi based systems [29] had the advantage that they used the, already implemented, WiFi mesh within the hospital [21], cutting down on upfront costs and effort. Some studies reported issues with managing battery cycles [16,25], which varied largely between two weeks and six months, depending on the technique and intensity of use. Due to poor marking, some personnel failed to be present in the specific registration area [28]. Furthermore, in some cases, it was reported that users took their tracker home [29] or lost it during their stay [29]. These were mostly elderly patients. Both issues could be mitigated by using scanning systems at building exits [29].

Depending on the system, upfront costs can vary largely, and they depend on the scale of implementation. Studies using WiFi based systems reported a tag unit price between €60 and €70 [29]. As these tags could not be sterilized [29], a maintenance cost of about €0.01 had to be spared for disposable bags [29]. Supporting software composed a large amount of the budget for these systems with licensing costs, ranging between €18,327 [29] and €50,000 [25]. Reported upfront costs for proprietary standalone systems were $600 (€540) for each reader and $20 (€18) per tag [25]. The use of computerized tracking and identification systems was deemed safe against attacks, due to the use of time-switching identification numbers [16]. Another study reported an implementation cost, for a Bluetooth based beacons system, of $2.70 per square feet (€26.37/m^2^) of covered area [16]. Other costs such as accessories (e.g., lanyards at €0.30 each [29]), computing resources, and IT personnel had to be taken into account, but were not specified in the included records.

Another important aspect that needs to be taken into account is the computing power, as it is a crucial determinant for system response time. For RFID, system response time was defined as the time it took for the system to ascertain the presence of the user, after this user entered a room. A range of 10–30 s in response time has been found for RFID [21,24]. Position changes in a room itself were detected in 30–60 s [24].

The implementation of active RFID technology in one study [28] led to a 61.0% decrease in medication dispensing time compared to conventional barcode identification.

Active RFID can be used to find the exact location of people (wearing the RFID beacon or tag) and can be used as an identification method. This can be done by detection via a reader. This means that identification and finding the exact spot of a person with RFID does not require extra actions from the user.

#### 3.1.2. Passive RFID

Only one article on passive RFID was included (Table 2). The study demonstrated that wearing a passive RFID tag during an MRI scan may cause small artifacts of 2–4 mm on the image. Furthermore, the MRI could cause the RFID tag to heat with a maximum of 3.6 °C and there might occur a movement of 1 N/kg at most. Neither after a CT scan, nor an MRI scan was there loss of memory or data alteration of the RFID tags [34].

Passive RFID can be used as an identification method, but it can also be used to locate a person. However, to find the location of a person, the user has to hold the passive RFID close to the reader actively. This means the location can only be found when the RFID is held close to the reader.

#### 3.1.3. Unspecified RFID

In three articles, it was not clear whether passive RFID tags or active RFID tags were used (Table 2). The recorded accuracy differed greatly between studies. In the study from Fisher et al., different hospitals were observed and interviewed. There were two hospitals that often could not find the tag, four hospitals that could, at least, find their tags inside the hospital, and one hospital that could even locate the tags with room-level accuracy [19]. In Saito et al., it was found that the position of the tag on the patient did not interfere with the accuracy provided, which was more than 95%. Furthermore, two studies recorded that there was no interference measured with other medical devices [33,35]. Ting et al. specified that tag usability depended on whether staff was trained to work with RFID and whether there was a focus on change management. It was also observed that 12% of the patients, who received a card with an RFID tag, forgot their card at a next visit [35].

In this section, it was not clear whether active or passive RFID was used. However, both active and passive RFID can be used to find the location of the users, and they can be used as identification methods.

### 3.2. Biometric

There were five articles that described the implementation or prototype testing of biometric identification techniques in a medical setting (Table 3).

#### 3.2.1. Fingerprints

Three articles reported fingerprints as a unique identifier, for matching bio- and demo-graphic data of an individual to their healthcare records, in low-economic resource-constrained global healthcare settings. White et al. brought up the limitation that fingerprinting cannot be used to identify persons < 5 years old. They also reported a failure of hardware and software in a quarter of the identification procedures [37]. Fingerprinting on average took 30s (n = 17,448) [36]. Wall et al. described a specificity and sensitivity ranging from 95–100% depending on the amount of fingers being printed. With both the thumb and index finger printed, the false matching rate was 1/1000 and the false rejection rate <1/10.000. The false matching rate was the issuance of an already existent ID to a new patient, and the false rejection rate was the failure to recall an existing ID [36]. Odei-Lartey et al. reported a sensitivity of 65.7% and a specificity of 100% in individuals >13 years old. Additionally, it showed a decline to a sensitivity below 15% for individuals ≤13 years old [27]. Two articles reported the value of adding extra information in the form of control questions (year of birth, name of father, etc.) or extra personal data (gender, date of birth, address, etc.) to lower the false matching rate [27,37]. While in the study of Odel-Lartey et al., none of the residents refused to give their fingerprint and photo (probably due to comprehensive information supply about the technique), privacy issues were a reason for refusal for half of the female sex workers [37].

#### 3.2.2. Facial Recognition

One article about facial recognition met the inclusion criteria. Facial recognition identifies users through landmarking of unique facial traits. This technique had a response time of 0.5 s when there were optimal light conditions. However if the light was too bright, too dark, or under a different angle, therefore creating shade, response time would increase up to 5 min. Another limitation was that partially covered faces could not be verified. The technique showed a sensitivity and specificity of respectively 99.7% and 100% (62 patients, 286 verifications) [23].

#### 3.2.3. Iris Scan

The study by Anne et al. explored the possibility of an iris recognition system [15]. Iris recognition is based on features such as the striation pattern, contrast ratio (between sclera and surrounding skin tone), and the difference between the right and left eyes. Eight-thousand eight-hundred ninety-four patients were approached, and one-hundred thirteen patients refused iris scans because of privacy issues, cultural/religious concerns, fear of the camera, or fear of exacerbation of an existing eye disease. In 5.3% of the cases, the system failed in its process to generate an ID, due to software or hardware issues or an eye deformity. The system had a sensitivity of 96.7%. The false match rate was 0.5%, and the false rejection rate was 4.8%. The response time was 20 s.

Biometric technology is mainly used as an identification method. For identification, some biometric technologies (fingerprint and iris) require the user to stand still and hold a certain body part in front of the scanner for a certain time. This is also the case with some facial recognition techniques. However, some facial recognition techniques do not require the user to stand still in front of a camera, but the user can move freely in a certain area while the camera is used to identify him/her. This last technique does not require extra actions from the user for identification. Biometric technology could also be used to find someone’s location. With respect to identification, someone’s location can be extracted.

### 3.3. Others

Two other records were included that described RTLS [19] using IR and US technology [22] (Table 4). Jeong et al. found that the rate at which IR badges were detected by a sensor was 95.6% [22]. Another article stated that locating IR badges were accurate enough to detect in which room the badges were present. Systems using US could be located inside the hospital; however, room-level accuracy was not reached [19].

Both IR and US technology could be used for identification and localization of the user. Neither technique required extra actions from the user; the badges only had to be read by a reader.

## 4. Discussion

In this qualitative systematic review, we gave an overview of the different real-time person identification methods in healthcare settings. The accuracy of active RFID varied between 52.4 and 100% [24,25,31]. The lowest accuracy of 52.4% was reached in a real-life setting, where healthcare workers were placed improperly before the sensor or they passed the sensor too quickly [31]. Overestimation of RFID performance in clinical practice is something to take into account in interpreting the results from test settings.

A problem that came up multiple times was the false appointment of a person to a location, because his/her badge was detected by a scanner with a large detection area, while the person was not physically in this particular location, but somewhere nearby. The detection area was determined by the power transmitted by the scanner [38], the number of scanners, and their physical distribution across a room or building. Determining the value of these parameters should be undertaken prior to implementation [39].

Most articles about biometric systems took place in lower socio-economic countries with less developed healthcare information technology, WiFi, and electricity networks compared to Western hospitals. The general accuracy of biometric systems was quite high; only fingerprint recognition had a broad range in sensitivity: 65.7–95% [27,36]. The relatively low sensitivity (65.7%) could be due to the poor quality of fingerprints. Studies were in conflict with whether this may be caused by the fact that some participants were engaged in farming or manual work and therefore had rough hands [27,37]. Response time to identify people entering a room in RFID was 10–30 s [21,24]. Identification by biometrics lasted on average 20 s [27] and 30 s [26] in iris recognition and fingerprinting, respectively. Facial recognition took 0.5 s to 5 min depending on the lighting conditions [23]. This relatively long response time could cause problems when quick identification was required, for example in emergency situations.

Substantial information about the different costs of RFID was found. These were composed of hardware, software, and IT maintenance costs and were a result of the extensiveness of the actual system. Unfortunately, information about the costs of biometric identification, US, or IR technology was not found.

### 4.1. Limitations of the Included Studies and Outcomes

Most of the reviewed articles discussed active RFID. Passive RFID tags, IR, US, and biometric identification methods such as facial, finger, and iris recognition were less frequently discussed. Active tags might be used and therefore were reported more frequently in comparison to passive RFID because they had an internal power source and could have a range up to 100 m, which made them more suitable for person identification [24]. This resulted in a more complete overview of active RFID tags. Articles that did not specify the type of RFID identification were valued less during the quality assessment due to the fact that the method of the article was not clear enough.

Next, RFID is a complex technique for identification or tracking consisting of multiple components such as badges, beacons, receivers/readers and antennas. In this review, the results were divided into three groups: active, passive, and unspecified. Hereby, we did not extensively explore differences in the various system components and mutual configuration, while these are important determinants for system performance.

Furthermore, some of the systems that were discussed in the articles contained a much broader application than just identification. Some articles made clear divisions between different components [19,20], while others did not [25,28]. Next to this, the definition of accuracy was not always specified, so it could be interpreted as tracing or linking medication to patients [28]. Moreover, it was assumed that RTLS techniques had an identification component, whilst, for example, only the amount of people at a specific location was tracked without identifying the individuals.

### 4.2. Limitations of the Review Method

Our search query resulted in multiple techniques, yet only a few fit the scope of the review. The articles that were excluded had no medical setting or no outcome measures. Due to these exclusion criteria, many valuable technical articles with specific outcome measures may have been lost. For example, many articles discussed the outcome “privacy” by proposing a solution (algorithm or scheme) for privacy problems. However, since they did not specifically describe an identification technique, but only an improvement, and they were not tested in a medical setting, they were excluded. The review conducted by Peteriya et al. gave an overview of safety and privacy issues concerning RFID and algorithms and authentication schemes that had been designed to conquer these problems [40]. Other outcomes such as cost, hygiene, and user safety were also not highly represented in the set of articles. This could also be the cause of these exclusion criteria or because they were just not frequently researched. Furthermore, many articles with new techniques that would be suitable in a medical setting, but were not yet tested or implemented, were lost. Examples of such techniques that were described in these articles were finger knuckle print and ear biometrics [41,42].

The quality and bias assessment were designed by the authors. The quality assessment was based on components that the authors found important for this subject (studies about identification methods in healthcare). The bias assessment was a small assessment where only a few types of biases were included. This meant both the quality and bias assessments were not standardized assessments.

### 4.3. Contribution and Future Research

There are many reviews about specific types of identification methods such as RFID [7]. However, little research has been done on a broader range of technical solutions for real-time person identification available for hospital application already implemented elsewhere. A few recommendations can be made based on the current results in light of the earlier named project that initiated this study. As of the publication of this paper, an initial design for the system mentioned in the introduction has been delivered. In this case, passive RFID in the form of NFC cards has been chosen as the preferred identification method. This choice was encouraged by the fact that this technique was already available in the hospital for which the system was intended. Furthermore, some legal issues weighted strongly against the use of biometric identification, not even accounting for costs and fluctuating accuracy. Concerning costs, this was a strong argument against active RFID in this specific use case. Active tags would have to be purchased and maintained for a pool of 200 healthcare workers for the single purpose of checking into the trauma room. Lastly, US and IR were left out of consideration as they appeared experimental and lacked documentation. Of course, every choice will be very context bound. However, strong recommendations can be made to observe data privacy laws of one’s country, as well as personnel opinion and cost benefit considerations. The results of this review could now be used as a guideline to choose a type of identification method for multiple medical applications, such as the trauma room stated in the Introduction. It is important to notice that this review reported the identification of both personnel and patients (Table 1). For further research, it is useful to answer the same question with a different approach. It would be advisable to create two search queries: (a) for the different techniques tested or implemented in a medical setting and (b) to find the technical specifications and the outcome measures for the techniques in (a). This could result in a better and more complete overview of all the outcome measures. Additionally, it would be advisable to include techniques that are applicable in medical settings even though they have not been tested or implemented.

## 5. Conclusions

In this systematic review, a variety of real-time person identification techniques in healthcare setting were found in the included studies. Although most of these studies were about active RFID tags, there were also studies about identification using passive or unspecified RFID tags, US, IR, facial, fingerprint, and iris recognition. RFID was mostly seen in hospitals in well developed countries, while biometrics were mostly seen in a global health setting. Substantial information was found about the accuracy, costs, and usability of active RFID; however, privacy and user safety were characteristics that were underrepresented in the results. Furthermore, a considerable amount of information was found about the accuracy, usability, and response time of biometric systems, including facial, fingerprint, and iris recognition. Little information was available for IR or US techniques and passive RFID tags.

To conclude, active RFID systems are already widely used in hospital settings. Therefore, much information can be found in the literature on this topic. Other techniques are still developing and have not been implemented or tested in hospital settings as much. This review provides an overview of real-time person identification systems that are already implemented or tested in hospital settings.

## Figures and Tables

**Figure 1 sensors-20-03937-f001:**
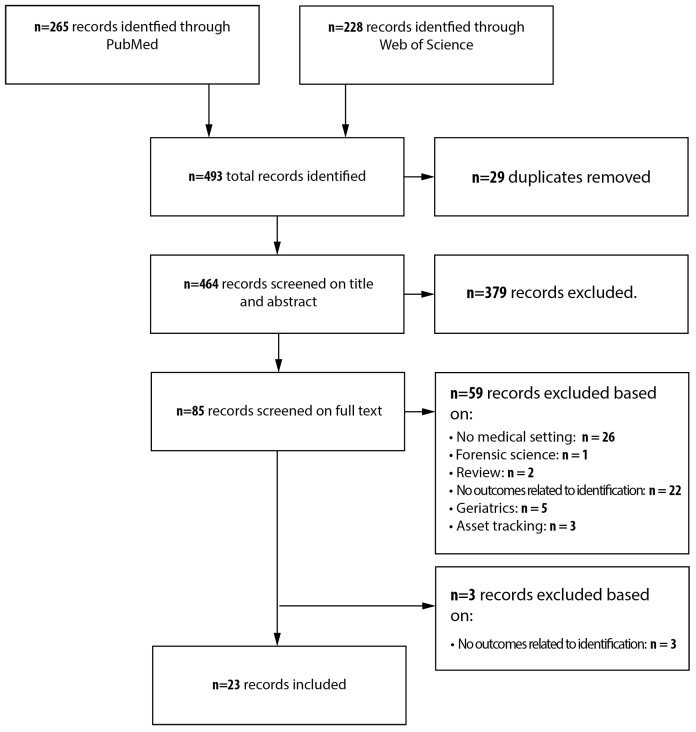
Flowchart of the inclusion process according to PRISMA.

**Table 1 sensors-20-03937-t001:** Overview of the included articles. Sample size and population refer to the test setting. The used identification technique, its technical specifications, and its application are outlined. ${}^{a}$/${}^{b}$: the scales used for the assessment of quality and bias are described in Appendix B.

Authors	Year	Design	Sample Size	Sample Characteristics	Quality Assessment *^a^*	Bias *^b^*	Technique/System	Specifications	Application
Anne et al. [15]	2020	Longitudinal study	8794	Patients	***	-	Iris scanner	Binocular iris recognition cameras (SMITech model BMT-20)	Patient identification for routine HIV program data for surveillance
Cao et al. [16]	2014	Case study	13	IT personnel	**	B	Active RFID tags	Battery-powered fixed receivers; mobile, battery-powered RFID beacons placed on badges	Personnel RTLS
Chang et al. [17]	2011	Pilot study	n.a.	n.a.	**	B	Active RFID tags	Four active RFID tags (125 kHz) and two tag readers	Identification of ICU staff to trace contact history of caregivers at the ICU with patients
Chen et al. [18]	2013	Pilot study	n.a.	n.a.	**	B	Active RFID tags	Active RFID with far-field communication (UHF 865–928 MHZ) with compact readers	Patient identification for tracking during hospital stay
Fisher et al. [19]	2012	Qualitative study	80 interviews and 23 hospitals	Interviews and hospitals	**	B	RTLS	RFID, WiFi, Wireless Local Area Network (WLAN), Ultra-Wide Band (UWB), infrared (IR), Zig-Bee, Bluetooth, or ultrasound (US)	Patient identification/tracking or personnel tracking in a.o.surgery, delivering medicine, and general hospital setting
Frisby et al. [20]	2016	Cross-sectional study	n.a.	n.a.	***	-	Active RFID tags	Active RFID tags on badges (Bluetooth low energy beacon) and Raspberry Pi in rooms	Personnel attendance to patients to compute door to doctor time at the emergency department
Hsu et al. [21]	2016	Cross-sectional study	3 per test	1 patient, 2 healthcare workers	*	A*	Active RFID tags	Active RFID tags with 3 active antennas	Location confirmation by RTLS to authorize X-ray use
Jeong et al. [22]	2017	Criterion validation study	25	25 neuroscience patients	**	B	Infrared (IR) transmitting badges	Infrared (IR) transmitting badges that are detected by ceiling sensors	Real-time location tracking for patients during 2 min walking test in the Neuroscience Acute Care/Brain Rescue Unit
Jeon et al. [23]	2019	Case study	30	Patients	***	B	Face recognition	Self-developed app on smartphone with external database	Patient identification throughout hospital stay
Kranzfelder et al. [24]	2012	Preclinical evaluation	6	3 surgeons, 3 engineers	**	B, C*	Active RFID tags	Active RFID transponder badges (2.45 GHz) with three sector antennas and one RFID sector controller	Position monitoring team members in the operating room
Lin et al. [25]	2012	Case study	20	medical staff	**	B	Active RFID tags	Active RFID tags (433 MHz) in a garment, one active antenna in the room	Personnel count for air filtration optimization in the operating room
Liu et al. [26]	2011	Pilot study	Test: n.a.; survey: 174	n.a. 56 surgeons, 41 anesthesia and recovery room nurses, 26 operative room and instrument room nurses, 30 staff of the ED	***	C*	Active RFID tags	Active RFID wristbands (2.4 GHz) with 80m transmission and RFID readers on the ceilings	Patient identification to control the workflow for surgical patients in the operation theater
Odei-Lartey et al. [27]	2016	Cross-sectional study	n.a.	n.a.	***	B	Fingerprint recognition	Hamster plus IV, SecuGen Inc.	Identification and registration of entering patients in a rural African setting.
Ohashi et al. [28]	2010	Feasibility study	5	Nurses and people pretending to be patients	**	A*, B	Active RFID tags	RFID Power Tag from Matrix Inc. (300 MHz), with a maximum communication distance of 3000 mm	A system using RFID for reducing misidentifications of patients in a smart hospital at the University in Tokyo
Pérez et al. [29]	Nov. 2016	Cross-sectional study	n.a.	n.a.	***	B	Active RFID tags	WiFi Active Aeroscout T2	Patient identification throughout hospital for safer medication matching
Pérez et al. [30]	Aug. 2016	Case study	n.a.	n.a.	***	B	Active RFID tags	WiFi Active Aeroscout tags	Patient tracking through hospital for efficient medication supply and safer medication matching
Pineles et al. [31]	2014	Pilot study	n.a.	n.a.	**	B, C	Active RFID tags	Active RFID badges	Presence detection in front of soap dispenser
Polycarpou et al. [32]	2012	Observational study	n.a.	Patients in the ward	**	B	Active RFID tags	Class 1 Generation 2 USB stick-like UHF RFID badges and wristbands	Patient identification in a hospital environment
Saito et al. [33]	2013	Case study	20 tests with 1–4 users	Lab personnel	*	A*, B, C*	RFID	RFID tags (953 MHz UHF) in a garment combined with one active antenna per room	Presence detection in the lab
Steffen et al. [34]	2010	Cross-sectional study	n.a.	volunteers	**	A*, B, C	Passive RFID tags	Copper etched and aluminum etched RFID tags	Identification of patients after MRI or CT scanning
Ting et al. [35]	2011	Exploratory case study	Test: 10; survey: unknown	None	**	A*, B	RFID	n.a.	Implementation of RFID with a patient identification system in a healthcare company
Wall et al. [36]	2015	Cross-sectional study	120, 42	Staff members, female sex workers	**	B, C	Fingerprint recognition	n.a.	Identification of female sex workers for HIV treatment
White at al. [37]	2018	Parallel, convergent study	919	Patients	***	B	Fingerprint reader	Optical fingerprint reader	Patient identification in a tuberculosis clinic

**Table 2 sensors-20-03937-t002:** Overview of the characteristics regarding costs, usability, accuracy, response time, hygiene, privacy, and user safety of the different RFID techniques (active, passive, and unspecified).

Type	Cost	Usability	Accuracy	Response Time	Hygiene	Privacy	User Safety
Active	€60–€70/tag AeroScout [30]	Patients should be reminded to take tags back to the hospital [30]	1–4 m accuracy in patient localization [21,30]	Detection by surveillance sector antenna within 30 s [24]	Non-sterilizable with an autoclave; should be packaged in a single use bag [29]	Less vulnerable to attacks by using a unique patient ID that changes over time [16]	-
	$2.7/sqft (€26.37/m^2^) beacon for RTLS [16]	Older patients can lose tags [29]	Calculated entering times into room accurate to 1 s [20]	Detection of position change between 30 and 60 s [24]			
	$600 (€545.31)/433 MHz reader [25]	Management of low battery in tags (collection and change) [30]	Not accurate if not worn on visible places [29]	Locating process within 20 s [21]			
	$20 (€18.18)/433 MHz tag [25]	Configuration first time use (battery test, number of channels, frequency of transmission) [30]	Wrist band only detected within 5 cm of the detector [32]	Reduction of 61% of medication dispensing time compared to a regular barcode based workflow [28]			
	€0.001 plastic bag for hygiene [30]	Battery life 2 weeks–6 months [16,25]	RFID tags attached to personal ID cards detected within 80 cm of the detector [32]				
	€0.3 lanyards [30]	WiFi infrastructure usually already present in hospitals [21]	Detection accuracy of 52.4% [31], 82.0% [17], 85.0% [16], 98.0% [25], 100% [24]				
	€18,327.11 [30]–€50,000 [25] for RTLS software	Registration area (range) has to be clearly marked [28]	Accuracy may differ by 10% between 1 and 2 readers in the room [25]				
		91.3% agreed that the system was conducive to improving patient identification [26]	Large influence of tag position on accuracy (20% decrease) [31]				
			False positives due to proximity [16,20,25,26]				
			Failure to detect when in too large or crowded areas [18]				
			1 hospital: able to locate tag, but no room-level accuracy [19]				
Passive	-	Showed small artifacts of 2–4 mm on MRI image [34]	-	-	-	-	- Little to no interaction in MRI [34]
		No memory loss or data alteration of RFID tags after MRI/CT scanning [34]					
Unspecified	-	12% of patients forget membership card with tag [35]	Location estimated accuracy 95% [33]	-	-	-	No interference with other medical devices in cardiac intervention lab [33]
		Usability depends on staff training and change management [35]	Accuracy independent of tag position on patient [33]				No interference with other medical devices at a distance >30 cm apart [35]
			2 hospitals: tag not found [19]				
			4 hospitals: able to locate tag, but no room-level accuracy [19]				
			1 hospital: room-level accuracy [19]				

**Table 3 sensors-20-03937-t003:** Overview of the characteristics regarding costs, usability, accuracy, response time, hygiene, privacy, and user safety of the different biometric techniques (facial, finger, and iris recognition).

Type	Cost	Usability	Accuracy	Response Time	Hygiene	Privacy	User Safety
Facial recognition	-	-	Sensitivity 99.7% and specificity 99.99% [23]	Verification could take from 0.5 s up to 5 min depending on lightning conditions [23]	-	-	-
			(Partially) covered faces could not be detected [23]				
Fingerprint recognition	-	26% failure of the technique in capturing fingerprints [37]	Sensitivity of 65.7% [27] to 95% [36] and 100% specificity [15,25]	Average reading time 30 s [27]	-	Full acceptance rate (100%) if correctly informed [27]	-
		Unable to capture individuals ≤ 5y old [37]	Thumb and index finger more accurate than index finger alone [36]			50% refusal for fingerprinting based on privacy issues [37]	
		Sensitivity <15% when capturing individuals ≤ 12y old [27]	False fingerprint matching 0.1% [36]				
Iris recognition	-	5.3% failure of technique in generating iris template or unique ID [15]	Sensitivity 94.7% [15]	Average identification time 20 s [15]	-	Technique is not used in any civil or governmental processes [15]	-
			False match rate 0.5% [15]			1% refusal rate due to privacy concerns [15]	
			False rejection rate 4.8% [15]				

**Table 4 sensors-20-03937-t004:** Overview of the characteristics regarding costs, usability, accuracy, response time, hygiene, privacy, and user safety of the category “other” techniques (IR and US).

Type	Cost	Usability	Accuracy	Response Time	Hygiene	Privacy	User Safety
Infrared (IR)	-	-	4.4% non-detection rate for IR based RTLS [22]	-	-	-	-
			Detection rate: 96% [22]				
			1 hospital: room-level accuracy [19]				
Ultrasound (US)	-	-	2 hospitals: able to locate tag, but no room-level accuracy [19]	-	-	-	-

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
