# Peer review of "Real-Time Person Identification in a Hospital Setting: A Systematic Review"

_sensors, 2020, doi:10.3390/s20143937_

Round 1

Reviewer 1 Report

The manuscript presents a systematic review of identification techniques used in medical settings. The methodology (including appendices) is logical, well documented, and reproducible for future studies. The findings present 7 techniques, emphasizing the bulk of reviewed studies focus on active RFID. Apart from a few minor typos, the manuscript is clearly written, free of jargon, and understandable to professional and lay readers. The Tables are very well organized and clear. It is unclear what the utility and implications of the quality/bias measures are that are included in the Tables and in Appendix B. This should be addressed in the discussion section. Otherwise, I have only minor comments on points of clarity and typos.

Abstract

p.1, lines 12-13: Originally I was confused by the seven techniques as the letter designations (a,b,c) follow rather than precede the techniques, which is the convention I’m aware of. Recommend using the convention of putting letters before the techniques as was done in the methods on p. 2, lines 52-53.

Methods

p.2, line 43: Recommend inserting “The” before “most important terms in our searchstring…”

Results

p.8, line 100: I believe “common” should be “commonly”? Also, “first and foremost” implies that something is the most important thing. Do the authors mean that RFID active is the most important group or do they simply mean it is the most common? If the latter, then first and foremost should be omitted.

p.8, line 123: After second reporting of citation [29]. Recommend period to start new sentence instead of semicolon.

p.11, lines 183-4: The sentence that begins “However when the light was too bright” is awkward as it difficult to interpret after which comma the list ends and the statements begins. Recommend revising for clarity.

p.11, line 188: Use of the word “of” in “They study of Anne et al” should use “by” as Anne et al were no the subjects of the study.

Discussion

p.12, line 209: Sentence “because its badge is detected” recommend using pronoun “their” instead of “its”

p.13, line 222: Recommend use of “relatively” instead of “relative”

p.13, line 238 and line 268: Recommend colon to introduce lists instead of semicolon

Author Response

Dear reviewer,

Let us first thank you for your critical reading of our work. We were pleased to read the encouraging remarks. We were especially careful to document the methodology as this was a tedious part of the process for us: we are pleased to hear this was not in vain. We shall start with addressing the small textual remarks:

  • Remark: letter designations in the abstract were put in front of the intended technique. (Line 12-13)
  • Remark: “The” was put in front of the sentence to resolve any ambiguity. (Line 47)
  • Remark: “commonly” has replaced “common” (Line 104)
  • Remark: “First and foremost” was replaced by “most” (Line 104)
  • Remark: semicolon replaced. (Line 126)
  • Remark: Sentences was reformulated to: “However if the light was too bright, too dark or under a different angle, therefore creating shade, response time would increase up to 5min.” (Lines 196-197)
  • Remark: “of” has been replaced with “by” (Line 201)
  • Remark: “their” has replaced “its” (Lines 230-231)
  • Remark: “Relatively” has replaced “relative” (Line 239)
  • Remark: semicolons have been replaced. (Lines 239 and 306)

We have added a short paragraph discussing the concerns and eventual implications of the quality/bias assessment (Lines 283 to 286). We hope this addressed the concerns you expressed in your review. We also added a paragraph in the introduction (Lines 31 to 34) to further contextualise our paper and allow for some recommendations at the end (Lines 290 to 301). Lastly, a few sentences were added on recommendation of another reviewer to clarify the distinction between real-time tracking and simple identification (Lines 147-150, 157-159, 171-172 and 208-214). 

Thank you again for your critical reading. Hopefully we replied in a satisfactory manner to your remarks, otherwise we will gladly review our manuscript.

Regards,

Nino Wouters on behalf of all listed authors.

Reviewer 2 Report

Thanks for the opportunity to review this interesting paper in an important area.  I'm a little concerned by the very small numbers of papers identified as being relevant. I think this may be partly because of the exclusion criteria, and whether there are techniques that are being used e.g. GPS and cellphone or other device tracking, that have the effect of tracking people.  I think it could also be helpful to emphasise the fact that these systems give both location and identification e.g. person x is at place Y. However some of them may locate people only in certain places e.g iris readers and some may give a continuous update of location e.g. RTLS. This could be another column in the tables - does the system constantly track or just  give location at specified places. The  degree to which user action is required may also be helpful - possibly more than hygiene or cost which are hard to assess for many systems. 

I think this review would be  more comprehensive if there was an attempt to expand the number of retrieved papers by adding some hand-searching, particularly in terms of technologies  which may be in reviews etc.  One of the issues that arises with systematic reviews of this type in my view is that the adherence to strict rules on search is entirely reasonable when a meta-analysis is being performed but when it is more of a review, "missing" technologies is more important.

Author Response

Dear Reviewer,

Thank you for the critical reading of our manuscript and sharing your concerns. The limited amount of included papers was indeed the result of a strict set of inclusion and exclusion criteria. The aim of this particular set of rules was to obtain a to the point assortment of papers that answered our question factually. Some techniques were beyond the scope of the question and not applicable to the context in which our study was performed. This context has been further explained in the revised paper following remarks from other reviewers. Furthermore, we agree fully with your suggestion concerning the distinction between RTLS and identification systems. We have tried to incorporate this suggestion in this revised version of the paper, both in the text as in the table. 

Finally, this review was performed as part of a graduation project and is obeying rules that required for a systematic study of literature. Manual search was therefore not performed. We realise that despite all the care we have taken into performing the search, this may cause for an incomplete picture. We have tried to address this in the discussion. We hope that nevertheless, our research will contribute to a certain extend to the concerned field of study and stimulate interest of the readers.

Some minor changes however have been made to the paper. We have added a short paragraph discussing the concerns and eventual implications of the quality/bias assessment (Lines 283 to 286). We also added a paragraph in the introduction (Lines 31 to 34) to further contextualise our paper and allow for some recommendations at the end (Lines 290 to 301). Lastly, a few sentences were added on recommendation of another reviewer to clarify the distinction between real-time tracking and simple identification (Lines 147-150, 157-159, 171-172 and 208-214).

Thank you again for reviewing our work. We hope to have provided you with sufficient information, otherwise we would gladly address further concerns.

Regards,

Nino Wouters on behalf of all listed authors.

Reviewer 3 Report

Well done paper. I have only minor comments:

Line 23, I assume that is only an example of one identification, so say it: "For example, ..."

Line 41, define "Deduplication", it is not a word in the dictionary so you need to give it a brief definition. 

Line 117, did you mean time-taking? Not "time-staking"?

Line 227, ".. biometrics were mostly seen in a global health setting." Did you mean Global South settings? Why is that case? It would be interesting to discuss why countries in the Global South use biometrics. 

Suggestion: How about making some recommendations based on your review?

Author Response

Dear Reviewer,

Let us first thank you for your critical reading of our work. We are pleased to hear that it met the standards of academic publishing and will hopefully soon contribute to the field. We were admittedly a bit shy in making suggestions to the reader. Due to the context specific considerations that are of influence on the decision concerning the system, it was difficult to make relevant recommendations. However, following the first round of reviews and some recommendations from supervisors, the paper has further been contextualised within our research problem to allow for some recommendations. We will explain these further on, first we addressed the textual remarks.

  • “For example” was added. (Line 23)
  • “Deduplication” was replaced with “Removal of duplicates within the retrieved articles[...]” (Lines 44 to 45)
  • “time-staking” was replaced with “time-consuming” your suggestion was correct nevertheless. (Line 121)
  • In this context, “global health” refers to the use of the technique in monitoring population groups, not per-se in southern nor emerging countries yet it was coincidental that only African countries were subject in the referred papers. Hopefully leaving the sentence as-is has your approval after clarification. (Line 315)

As said, we have contextualised our paper in the project is was part of (Lines 31-34) which also allowed for some recommendations at the end (Lines 290 to 301). We have also added a short paragraph discussing the concerns and eventual implications of the quality/bias assessment (Lines 283 to 286). Lastly, a few sentences were added on recommendation of another reviewer to clarify the distinction between real-time tracking and simple identification (Lines 147-150, 157-159, 171-172 and 208-214).

We thank you again for your critical reading and suggestions to improve our work. We hope to have addressed all your concerns and would otherwise be glad to improve further on our work.

Regards,

Nino Wouters on behalf of all the listed authors.